



# Reanalysis of vertical mixing in mesocosm experiments: PeECE III and KOSMOS 2013

Sabine Mathesius[1], Julia Getzlaff[1], Heiner Dietze[1,2], Andreas Oschlies[1,2], and Markus Schartau[1]

[1]GEOMAR Helmholtz Centre for Ocean Research Kiel
[2]Kiel University (CAU)

*Correspondence to:* Sabine Mathesius (smathesius@geomar.de)

**Abstract.** Controlled manipulation of environmental conditions within large enclosures in the ocean, so-called pelagic mesocosms, has become a standard method to explore potential responses of marine plankton communities to anthropogenic change. Among the challenges of interpreting mesocosm data is the often uncertain role of vertical mixing, which usually is not observed directly. To account for mixing nonetheless, two pragmatic assumptions are common: either that the water column is homogeneously mixed or that it is divided into two water bodies with a horizontal barrier inhibiting turbulent exchange. In this study, we present a model-based reanalysis of vertical turbulent diffusion in the mesocosm experiments PeECE III and KOSMOS 2013. Our diffusivity estimates indicate intermittent mixing events along with stagnating periods and yield simulated temperature and salinity profiles that are consistent with the observations. Here, we provide the respective diffusivities as a comprehensive data product in the Network Common Data Format (NetCDF). This data product will help to guide forthcoming model studies that aim at deepening our understanding of biogeochemical processes in the PeECE III and KOSMOS 2013 mesocosms, such as the $CO_2$-related changes in marine carbon export. In addition, we make our model code available, providing an adjustable tool to simulate vertical mixing in any other pelagic mesocosm. The data product and the model code are available at doi: 10.1594/PANGAEA.905311 (Mathesius et al., 2019).

## 1 Introduction

In a rapidly changing world (Steffen et al., 2011, 2018; Stocker et al., 2014), a better understanding of anthropogenic pressures on marine ecosystems and biogeochemical cycles is needed, so that major risks can be identified in advance and mitigated. The response of marine plankton communities to anthropogenic change is often explored in so-called pelagic mesocosms, large polyethylene bags, either drifting or being mounted in coastal waters, usually 2 m in diameter and up to 20 m long. They are filled with local seawater and contain a natural plankton community (Riebesell et al., 2008, 2013). Mesocosms are big enough to prevent many 'bottle effects' that are existent in laboratory experiments, while at the same time they are small enough to allow for a cost-efficient controlled manipulation of environmental conditions and frequent monitoring of species composition and biogeochemistry. Therefore, they provide unique insights into possible responses of planktonic ecosystems to anthropogenic pressures that cannot be obtained from laboratory studies. Past mesocosm experiments investigated, e.g., the effects of decreasing pH (Riebesell et al., 2008, 2013; Gazeau et al., 2016; Engel et al., 2014; Galgani et al., 2014; Archer et al.,



2018), nutrient increase (Schwier et al., 2017; Micheli, 1999) and warming (Lewandowska et al., 2014; Sommer et al., 2015; Wohlers et al., 2009). Ocean acidification has been found to have a significant impact on organic matter stoichiometry (Riebesell et al., 2007; Schulz et al., 2008), plankton community composition (Bach et al., 2016; Riebesell et al., 2017; Boxhammer et al., 2018), primary production and carbon export (Egge et al., 2009; Riebesell et al., 2007). Even though mesocosms provide a

well-controlled environment, where causes and consequences are much easier to disentangle than in the field, uncertainties remain. The observed changes in biogeochemistry and community structure are usually a result of numerous unobserved processes, so there is often room for interpretation, with more than one plausible hypothesis regarding the key events that took place during the experiment. Models are a valuable complement to statistical data analyses, because they can explicitly resolve and investigate unobserved processes and therefore test different assumptions and hypotheses. In this study, we focus on the

uncertain role of vertical mixing, which is very hard or even impossible to measure directly without effectively mixing the water column by introducing respective measurement devices. So far, the interpretation of some of the most influential mesocosm experiments could not take vertical mixing into account, since data on mixing were not available. However, vertical mixing can be an important factor that affects sinking and resuspension of particulate matter, dilution and entrainment of nutrients, as well as air-sea $CO_2$ fluxes. Accounting for vertical mixing in the interpretation of mesocosm data might provide a better

understanding of the observed biogeochemical changes. To this end, we developed a one-dimensional mesocosm model that is able to retrace or reanalyse vertical turbulent diffusivities based on temperature, salinity and solar radiation data. These data sets are available for nearly all mesocosm experiments, so that this model can be used to simulate vertical mixing in almost any pelagic mesocosm. In this paper, we present simulations of the mesocosm experiments PeECE III (Pelagic Ecosystem $CO_2$ Enrichment III; Schulz et al., 2008) and KOSMOS 2013 (Kiel Off-Shore Mesocosms for Future Ocean Simulations 2013; Bach

et al., 2016). These experiments exposed marine plankton communities to elevated $CO_2$ levels and provided highly valuable insights on the plankton community response to ocean acidification. There are indications that the vertical mixing patterns in the mesocosm experiments PeECE III and KOSMOS 2013 are complex, since a variety of forcing mechanisms is at work. Among them (1) thermal stratification due to solar radiation, (2) a diurnal cycle in air-sea heat and associated buoyancy fluxes, where surface waters are cooled at night, inducing destabilization of the water column and convection, and (3) wind friction driving

shear and waves which impinge on the flexible mesocosm walls, thereby turbulently pushing water up and down. Despite evidence that substantial mixing as well as strong stratification has occurred in the mesocosms, quantitative estimates have not been available. Here, we publish the model code and our estimated turbulent diffusivities for the PeECE III and KOSMOS 2013 mesocosm experiments, available at doi: 10.1594/PANGAEA.905311 (Mathesius et al., 2019). Furthermore, we demonstrate that these estimates provide realistic profiles of temperature and salinity that are consistent with the observations. The mixing

model can easily be coupled to any plankton ecosystem mesocosm model (e.g., the model of Krishna and Schartau, 2017). In a follow-up study, we will present the effect of vertical mixing on biogeochemical tracers in mesocosms.



## 2 Methods

### 2.1 Data

In this paper, we show the simulations of two mesocosm experiments, PeECE III (Schulz et al., 2008; Riebesell et al., 2007) and KOSMOS 2013 (Bach et al., 2016). The experiments differ in duration, length of mesocosms, environmental conditions and initial conditions. At the beginning, the PeECE III mesocosms are strongly stratified, while the KOSMOS 2013 mesocosms are fully mixed and then get partially stratified during the course of the experiment. Thus, these two mesocosm experiments cover a spectrum of different mixing patterns and provide a good test ground for our mixing model.

The PeECE III experiment was conducted in the fjord of Bergen, Norway, from May $5^{th}$ to June $15^{th}$, 2005. Nine mesocosms of 8.5 m length were deployed in the fjord and sampled every day. Temperature and salinity profiles were measured daily by a CTD. Before the experiment started, fresh water was mixed into the upper 5.5 m to create an artificial stratification. Throughout the experiment the upper surface layer was mixed by a pump to guarantee a homogeneous distribution of dissolved compounds within the mixed layer. Solar radiation data were provided by the Geophysical Institute of the University of Bergen (Olseth et al., 2006). Data on chlorophyll$_a$ concentrations were provided via Pangaea by the PeECE III team (2008).

For the KOSMOS 2013 experiment, ten mesocosms of 19 m length were deployed in the Gullmar Fjord, located near Kristineberg at the west coast of Sweden, from March $7^{th}$ to June $28^{th}$, 2013. CTD measurements and depth-integrated water samples were taken every other day (data provided by Boxhammer et al., 2017). There was no artificial stratification, instead the water column was fully mixed at the start of the experiment. For our simulations, we used solar radiation data from the nearby location of Kristineberg, provided by the Sven Lovén Centre for Marine Infrastructure of the University of Gothenburg (http://www.weather.loven.gu.se/kristineberg/en/data.shtml). Data on chlorophyll$_a$ concentrations were provided via Pangaea by Boxhammer et al. (2017).

For our simulations and model-data comparisons, we linearly interpolate the CTD data to hourly values. Data gaps are filled by linear interpolation as well. Since temperatures were measured only at daytime, the diurnal cycle of temperature was not observed and nightly cooling cannot be resolved in our model (implications are discussed in Section 4.5).

### 2.2 Model Description

The model has a vertical resolution of 0.5 m. The temporal resolution is process-specific: one second for the turbulent diffusion, ten minutes for the convective adjustment and one hour for the radiative heating and the surface corrections (see below). As initial profiles for temperature and salinity in each mesocosm, we take the average of the measured temperature and salinity profiles of the first three experiment days.

To simulate the temporal evolution of temperature and salinity within the mesocosms, we set up a one-dimensional turbulent diffusion model for these two tracers. We start with the general diffusion equation for a tracer $\phi$, which is given by

$$\frac{\partial \phi}{\partial t} = \frac{\partial}{\partial z}\left(k_z \cdot \frac{\partial \phi}{\partial z}\right), \tag{1}$$





where $k_z$ denotes the depth-dependent eddy diffusivity and $z$ the depth. The diffusion equation is solved based on a code of Burkardt (2009) that uses second-order central differences to approximate the second derivative in space and an implicit Euler approximation for the first derivative in time.

The eddy diffusivity $k_z$ is parameterised based on Osborn (1980), as

$$k_z = \frac{c}{N_z^2}, \tag{2}$$

where the parameter $c$ corresponds to the product of dissipation rate and mixing efficiency of Osborn's original parametrisation, and $N_z^2$ is the buoyancy frequency given by

$$N_z^2 = \frac{g}{\rho_z} \cdot \frac{\partial \rho_z}{\partial z}, \tag{3}$$

where $g$ denotes the gravitational acceleration and $\rho_z$ the potential density of depth $z$ (with $z$ defined positive downward).
In general, the dissipation rate as well as the mixing efficiency can significantly vary in time and space. However, our model assumes that $c$ is constant for all depths and the whole time period of the experiment. The constant $c$, which is optimised for every single mesocosm independently (see 2.3), can be interpreted as an averaged value within one mesocosm for the entire experiment time. With this simplification, any differences in the vertical diffusivities $k_z$ are induced by the buoyancy frequency and are inversely proportional to the vertical density gradient. The density gradients and the diffusivities are calculated each
hour from the interpolated observed temperature and salinity profiles. By using the observed values instead of simulated temperature and salinity, we avoid follow-up errors that could grow rapidly. Simulated temperature and salinity profiles only become relevant for the diffusivity calculation when the associated simulated density gradient is negative. In this case, we prescribe that convection takes place and we set $k_z$ to a high value of 0.1 m$^2$s$^{-1}$, which is in general the maximum value that diffusivities can take in our model. We check every ten minutes if conditions for convective mixing are present.
Once per hour, we carry out a surface correction of temperature and salinity (Sections 2.2.2, 2.2.3) and account for heating by solar radiation (Section 2.2.1).

### 2.2.1  Radiative Heat Flux

Solar radiation penetrates the water column and warms subsurface waters. To calculate the warming related to the radiative heat flux, we use solar radiation observations (see 2.1) as incoming solar radiation.
Within each box of the model, the radiative heating rate (RHR) is calculated, following Ohlmann (2003), as

$$RHR_z = \frac{SI_{in} - SI_{out}}{c_p \cdot \rho_z \cdot \Delta z}, \tag{4}$$



where $SI_{in}$ denotes the solar radiation entering the box, $SI_{out}$ the solar radiation leaving the box, $\Delta z$ the box thickness (0.5 m) and $c_p$ the specific heat capacity of seawater ($c_p = 3990$ W s kg$^{-1}$ K $^{-1}$). The amount of solar radiation leaving the box is given by

$$SI_{out} = SI_{in} \cdot e^{-\alpha \Delta z}, \tag{5}$$

and depends on $SI_{in}$ and the absorption coefficient $\alpha$, given as

$$\alpha = \alpha_w + \alpha_{chl} \cdot chl, \tag{6}$$

where $\alpha_w$ is the absorption coefficient of water ($\alpha_w = 0.04$ m$^{-1}$), $\alpha_{chl}$ the absorption coefficient of chlorophyll$_a$ ($\alpha_{chl} = 0.01$ m$^2$ (mg $chl$)$^{-1}$) and $chl$ the chlorophyll$_a$ concentration. Here we use the chlorophyll$_a$ concentrations measured during the mesocosm experiments. If these were not available, modelled chlorophyll$_a$ concentrations could be used.

The product of $RHR_z$ and $\Delta t$ is the radiative heating per simulation time step and is added to the respective temperature at depth $z$.

### 2.2.2  Temperature Surface Correction

Apart from the radiative heat flux, the water temperature is strongly influenced by sensible and latent heat fluxes at the sea surface. In our model set-up we assume that all changes in total heat content within a mesocosm are caused by the combination

of radiative heat fluxes and sea-surface heat fluxes.

To account for the sea-surface heat fluxes, we determine the hourly changes in simulated heat content (Eq. 7) and hourly interpolated observed heat content. The difference in heat content change is converted back to temperature (Eq. 8) and added to the top box of the simulated mesocosm, thus adding missing heat or subtracting excess heat.

The water-column heat content, $H$ in units of J m$^{-2}$, is calculated each hour for observed and simulated heat, according to

$$H = c_p \cdot \int_{z=0}^{z_{max}} \rho_z \cdot T_z \; dz \tag{7}$$

where $\rho_z$ and $T_z$ are the potential density and the temperature at depth $z$, respectively. We call the hourly change in observed heat $\Delta \mathrm{H}_{obs}$ and the hourly change in simulated heat $\Delta \mathrm{H}_{sim}$. The surface temperature correction term $T_{corr}$, which accounts for both sensible and latent heat fluxes, is then given by

$$T_{corr} = \begin{cases} \frac{(\Delta H_{obs} - \Delta H_{sim})}{\rho_1 \cdot c_p \cdot \Delta z} & |\, z \leq 0.5\,\mathrm{m} \\ 0 & |\, z > 0.5\,\mathrm{m} \end{cases} \tag{8}$$





where $\rho_1$ is the potential density of the top box of the model.

The complete temperature equation that includes the turbulent thermal flux, radiative heating and the sea-surface heat flux correction is then given by

$$\frac{\partial T}{\partial t} = \frac{\partial}{\partial z}\left(k_z \cdot \frac{\partial T}{\partial z}\right) + RHR_z + \frac{\partial \tilde{T}_{corr}}{\partial t}, \tag{9}$$

5 where $\tilde{T}_{corr}$ represents the continuous form of $T_{corr}$.

### 2.2.3 Salinity Surface Correction

Similar to the heat correction above, we also introduce a correction term for salinity that corrects for sea-surface fluxes that are not explicitly resolved in the model. We assume that all changes in salinity are caused by freshwater fluxes at the surface, driven by evaporation or precipitation. Every hour, we add the hourly change of observed depth-integrated salinity, $S_{corr}$, to
10 the top box of the model.

$$S_{corr} = \begin{cases} \int_{z=0}^{z_{max}} \Delta S_{obs}\, dz & |\, z \leq 0.5\,\mathrm{m} \\ 0 & |\, z > 0.5\,\mathrm{m} \end{cases} \tag{10}$$

where $\Delta S_{obs}$ denotes the hourly change in observed salinity.

The complete salinity equation that includes the turbulent salinity flux and the sea-surface freshwater flux correction can be written as

15 $$\frac{\partial S}{\partial t} = \frac{\partial}{\partial z}\left(k_z \cdot \frac{\partial S}{\partial z}\right) + \frac{\partial \tilde{S}_{corr}}{\partial t}, \tag{11}$$

where $\tilde{S}_{corr}$ represents the continuous form of $S_{corr}$.

### 2.3 Optimisation of the Diffusivity Parameter

To find the best diffusivity estimates for each mesocosm, we optimise the parameter $c$ of Eq. 2, by minimizing three different cost functions that either regard i) temperature alone, ii) only salinity, or iii) both tracers together. For the optimisation
20 we take only the daily measurements into account, not the interpolated hourly values. Since the measurements were taken approximately at noon, we compare them to a three-hour average of the simulated tracer values around noon.

When optimising $c$ depending on temperature, the averaged cost function $J_{\mathrm{T}}$ is given by

$$J_{\mathrm{T}} = \frac{1}{2\sigma_{\mathrm{T}}^2 n} \sum_{i=1}^{n} (T_{obs_i} - T_{sim_i})^2, \tag{12}$$





where $T_{obs}$ and $T_{sim}$ denote the observed and simulated temperature, respectively, $\sigma_\mathrm{T}$ the standard deviation of observed temperature in the mesocosm and $n$ the number of data points (i.e., the product of the number of experiment days and depth levels).

When optimising $c$ depending on salinity, the averaged cost function $J_\mathrm{s}$ is given by

$$5 \quad J_\mathrm{s} = \frac{1}{2\sigma_\mathrm{s}^2 n} \sum_{i=1}^{n} (S_{obs_i} - S_{sim_i})^2, \tag{13}$$

where $S_{obs}$ and $S_{sim}$ denote the observed and simulated salinity, respectively, and $\sigma_\mathrm{s}$ the standard deviation of observed salinity in the mesocosm.

Furthermore, we optimise the diffusivity parameter $c$ for both temperature and salinity simultaneously. In this case the so-called combined cost function $J_\mathrm{TS}$ is given by

$$10 \quad J_\mathrm{TS} = J_\mathrm{T} + J_\mathrm{s}, \tag{14}$$

The cost functions $J_\mathrm{T}$, $J_\mathrm{s}$ and $J_\mathrm{TS}$ are computed for different values of $c$, ranging from $10^{-8.5}$ to $10^{-6.5}$ in logarithmic equidistant steps of 0.1. The optimal value of $c$ is where the cost function reaches a minimum. In the following, $c_\mathrm{T}$ denotes the optimal value for $c$ when the cost function is optimised with respect to temperature only (Eq. 12), $c_\mathrm{s}$ when the cost function is optimised with respect to salinity only (Eq. 13), and $c_\mathrm{TS}$ when the cost function is optimised with respect to temperature and salinity simultaneously (Eq. 14) . The respective costs are given by $J_\mathrm{T}(c_\mathrm{T})$, $J_\mathrm{s}(c_\mathrm{s})$ and $J_\mathrm{TS}(c_\mathrm{TS})$.

## 3 Results

### 3.1 PeECE III

First, we optimise the diffusivity parameter $c$ by finding the corresponding minimum cost. Figure 1 shows the cost values resulting from the temperature optimisation, salinity optimisation, and the optimisation that depends on both temperature and salinity. We find a distinct minimum for each optimisation and mesocosm. This minimum determines the optimal value for $c$ that we use to calculate the turbulent eddy diffusivities (Eq. 2). In all mesocosms, the best estimates of $c$ resulting from the temperature optimisation ($c_\mathrm{T}$) are significantly larger than the best estimates of the salinity optimisation and the optimisation that depends on temperature and salinity simultaneously ($c_\mathrm{s}$ and $c_\mathrm{TS}$, respectively).

Figure 2 shows the observed temperature and salinity of one exemplary mesocosm (#8) as well as the simulated temperature, salinity and diffusivity that result from simulations with the three optimisations' best estimates for $c$ ($c_\mathrm{s}$, $c_\mathrm{T}$, and $c_\mathrm{TS}$).

When optimising $c$ depending on salinity only (Fig. 2, second row) the resulting salinity profiles are very similar to the observations. In comparison, the deviation of simulated temperatures from observed temperatures is much larger, especially from day 16 onwards. When optimising $c$ depending on temperature only (Fig. 2, third row), the simulated temperature profiles



**Table 1.** Best estimates of parameter $c$ ($c_T$, $c_S$, $c_{TS}$) and associated combined cost that takes temperature and salinity errors into account, $J_{TS}(c_T)$, $J_{TS}(c_S)$ and $J_{TS}(c_{TS})$, for all mesocosms of the experiment PeECE III.

| mesocosm | $\log_{10}(c_T)$ | $J_{TS}(c_T)$ | $\log_{10}(c_S)$ | $J_{TS}(c_S)$ | $\log_{10}(c_{TS})$ | $J_{TS}(c_{TS})$ |
|---|---|---|---|---|---|---|
| 1 | -7.4 | 0.296 | -8.2 | 0.096 | -8.1 | 0.094 |
| 2 | -7.4 | 0.275 | -8.2 | 0.069 | -8.1 | 0.066 |
| 3 | -7.4 | 0.242 | -8.0 | 0.089 | -8.0 | 0.089 |
| 4 | -7.4 | 0.267 | -8.2 | 0.087 | -8.1 | 0.079 |
| 5 | -7.3 | 0.317 | -8.2 | 0.077 | -8.1 | 0.074 |
| 6 | -7.5 | 0.164 | -8.0 | 0.069 | -7.9 | 0.069 |
| 7 | -7.2 | 0.318 | -8.1 | 0.090 | -8.0 | 0.082 |
| 8 | -7.2 | 0.361 | -8.3 | 0.093 | -8.1 | 0.082 |
| 9 | -7.2 | 0.350 | -8.2 | 0.084 | -8.1 | 0.081 |

**Table 2.** Mean and standard deviation of observed temperature (T) and salinity (S) as well as root mean square errors (RMSE) of simulated temperature and salinity, based on the simulation with $c_{TS}$ as optimal value for parameter $c$. The listed values are for all mesocosms of the experiment PeECE III.

| mesocosm | $T_{obs}$ (mean $\pm$ sd) | RMSE(T) | $S_{obs}$ (mean $\pm$ sd) | RMSE(S) | $\log_{10}(c_{TS})$ |
|---|---|---|---|---|---|
| 1 | $9.97 \pm 0.79$ | 0.257 | $31.12 \pm 0.54$ | 0.155 | -8.1 |
| 2 | $9.95 \pm 0.81$ | 0.263 | $31.17 \pm 0.52$ | 0.087 | -8.1 |
| 3 | $9.94 \pm 0.77$ | 0.258 | $31.12 \pm 0.41$ | 0.105 | -8.0 |
| 4 | $9.98 \pm 0.82$ | 0.296 | $31.09 \pm 0.59$ | 0.100 | -8.1 |
| 5 | $9.94 \pm 0.76$ | 0.269 | $31.12 \pm 0.56$ | 0.086 | -8.1 |
| 6 | $9.97 \pm 0.77$ | 0.240 | $31.04 \pm 0.44$ | 0.090 | -7.9 |
| 7 | $9.97 \pm 0.77$ | 0.281 | $31.03 \pm 0.54$ | 0.095 | -8.0 |
| 8 | $9.93 \pm 0.77$ | 0.263 | $31.09 \pm 0.55$ | 0.120 | -8.1 |
| 9 | $9.97 \pm 0.78$ | 0.270 | $31.07 \pm 0.59$ | 0.122 | -8.1 |

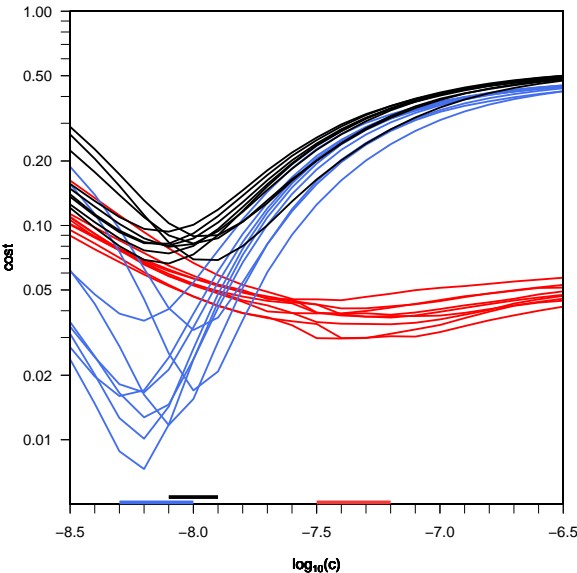

**Figure 1.** Costs of three cost functions depending on parameter $c$ (Eq. 2), for the experiment PeECE III. Black lines illustrate the cost function $J_{TS}$, taking into account the model-data misfit of temperature and salinity. Red lines illustrate the cost function $J_T$, taking into account only temperature errors; blue lines illustrate the cost function $J_S$, taking into account only salinity errors. The horizontal bars on the x-axis mark the range of all mesocosms' best estimates of $c$ for the three optimisations.

are very similar to the observations. However, there is some excess heat in the lower 2 m, especially after the surface heat peak. The simulated salinity pattern differs substantially from the observed pattern, starting already during the first four days of the simulation with a rapid weakening of the halocline.

When temperature and salinity are both taken into account in the optimisation (Fig. 2, bottom row), the results for temper-
ature, salinity and diffusivity are very similar to the results of the salinity optimisation. The simulated temperatures deviate more from the observations than in the temperature optimisation, whereas the simulated salinities are almost as accurate as in the salinity optimisation.

The reason for the higher impact of salinity is that, even though diffusivities resulting from a $c$ value higher than $c_{TS}$ improve temperature results, they also cause a quick dissolution of the halocline. This introduces large model errors and thus high costs
from the salinity term in Eq. 14. In the opposite case, diffusivities resulting from a $c$ value lower than $c_{TS}$ preserve the halocline well and at the same time lead to a temperature pattern that still resembles the basic characteristics of the observed pattern, so costs from the temperature term in Eq. 14 are relatively small. Thus, $c_{TS}$ is much closer to $c_S$ than to $c_T$.

To underpin these findings, we calculate the cost $J_{TS}$ depending on temperature and salinity for each optimised simulation. Table 1 shows the three optimisations' best estimates for parameter $c$ and the corresponding cost from the combined cost
function, $J_{TS}(c_T)$, $J_{TS}(c_S)$ and $J_{TS}(c_{TS})$, for each mesocosm. In general, $c_S$ and $c_{TS}$ are very close to each other, in one mesocosm

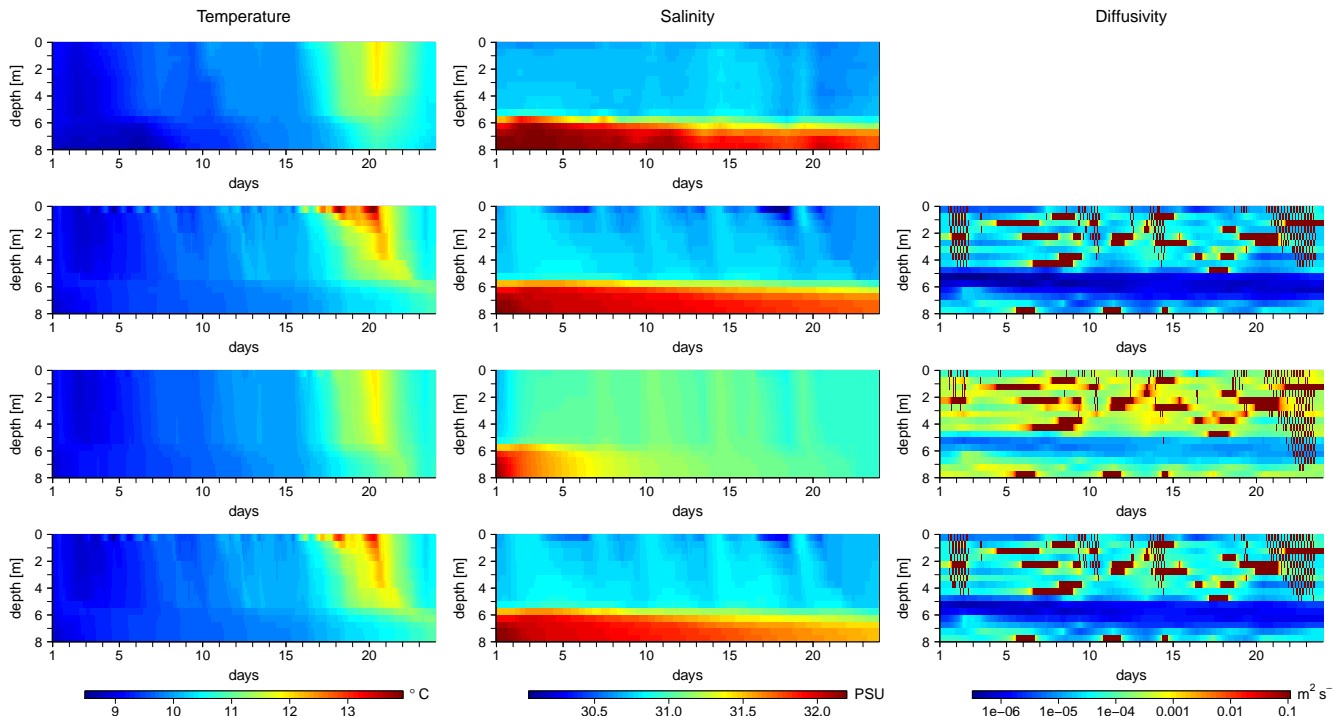

**Figure 2.** Temporal evolution of temperature, salinity, and diffusivity profiles in one exemplary mesocosm of PeECE III. The first row shows observed temperature and salinity; the following rows show simulated temperature, salinity, and diffusivity for a) the salinity optimisation (second row), b) the temperature optimisation (third row), and c) the optimisation that depends on both temperature and salinity (fourth row). Shown are results for mesocosm #8, figures for the other PeECE III mesocosms are provided in the Supplementary Material.

even identical, whereas all mesocosms' estimated $c_T$ are substantially higher. The cost values $J_{TS}(c_T)$ are in all mesocosms larger than the cost values $J_{TS}(c_S)$. The combined cost resulting from $c_{TS}$, $J_{TS}(c_{TS})$, provides the lowest cost. Thus, the model yields the best results when both temperature and salinity are included in the optimisation of diffusivities.

The optimised model simulations are further evaluated by a comparison of the root mean square error (RMSE). Here we focus on the best fitted simulation resulting from the simultaneous temperature and salinity optimisation. For each mesocosm, Table 2 shows mean and standard deviation of observed temperature and salinity as well as the RMSE of simulated temperature and salinity. The mesocosms' mean temperatures range between 9.93 and 9.98°C with a standard deviation of 0.76-0.82°C, while the mean salinities range between 31.03 and 31.17 PSU with a standard deviation of 0.41-0.59 PSU. The RMSE of temperature ranges from 0.24 to 0.30°C, whereas the RMSE of salinity ranges from 0.09 to 0.15 PSU. The highest RMSE of temperature and salinity in all mesocosms of the PeECE III experiment constitutes only 36.6% and 28.5% of the standard deviation of temperature and salinity observations, respectively.



## 3.2   KOSMOS 2013

For the KOSMOS 2013 experiment, we determine the optimal diffusivities by performing three optimisations, depending on either temperature or salinity only, or depending on both (see Fig. 3). The resulting cost values for each parameter variation are quite different from the costs of the previous experiment.

In the salinity optimisation, the cost minima are not as pronounced as they are in PeECE III. Best estimates of $c_s$ are higher for all KOSMOS 2013 mesocosms, if compared with those of the PeECE III experiment. Furthermore, the range of $c_s$ values is substantially larger. The opposite is the case in the temperature optimisation, where the cost minima are very pronounced, the range of $c_T$ is narrower and the cost curves of all mesocosms are almost identical. The $c_{TS}$ values are much higher than in PeECE III, thereby partly overlapping with the estimated $c_T$ and $c_s$ values.

Table 3 lists the best estimates for parameter $c$ ($c_T$, $c_s$ and $c_{TS}$) and the corresponding cost $J_{TS}$ for all mesocosms. In four mesocosms, $c_T$ equals $c_{TS}$, i.e. the temperature optimisation provides the same optimal $c$ value and therefore the same diffusivities as the simultaneous temperature and salinity optimisation. In two mesocosms, $c_s$ equals $c_{TS}$. In the remaining mesocosms, $c_{TS}$ is between $c_T$ and $c_s$ and provides the lowest cost. When $c_{TS}$ equals $c_s$ or $c_T$, the corresponding combined cost values are identical. However, whenever the optimal $c$ values are not identical, the simultaneous temperature and salinity optimisation

leads to the lowest model-data misfit.

Figure 4 shows the observed temperature and salinity of one exemplary mesocosm (#10) as well as the simulated temperature, salinity and diffusivity that result from simulations with the three optimisations' best estimates for $c$. The depicted mesocosm (#10) is the one that shows the largest spread between $c_s$, $c_T$ and $c_{TS}$.

The observed temperature and salinity profiles are substantially different from the conditions found in the PeECE III exper-
iment. Within the first five weeks, the whole water column is well mixed and characterized by temperatures below 4.8°C and salinities below 29.5 PSU. Over the course of the experiment, temperature and salinity increase to up to 16.8 °C and 29.66 PSU, respectively, and a pronounced thermal stratification is established for several weeks.

In all three optimisations, the model is able to reproduce the observed features very well. Notably, the simulated temperature and salinity profiles of the simultaneous temperature and salinity optimisation are more similar to the profiles of the temperature
optimisation, whereas in PeECE III they were closer to the profiles of the salinity optimisation.

The diffusivities resulting from the simulations with $c_T$ and $c_{TS}$ are very similar as well, while the diffusivities of the salinity optimisation are significantly larger. The right column of Fig. 4 shows that the simulated water column is well mixed during the first five weeks, followed by an increasingly distinct stratification with lower diffusivities between surface and bottom water masses. However, there are periods of deep convective mixing, reaching almost down to the mesocosm bottom. Like in the
optimised simulations of PeECE III, heat at the bottom is slightly overestimated in all simulations.

The mesocosms mean temperatures range between 7.36 and 7.40°C with a standard deviation of 4.49-4.53°C, while the mean salinities range between 29.10 and 29.45 PSU with a standard deviation of 0.07-0.11 PSU (Tab. 4). We further evaluate the optimised model simulations by a comparison of the root mean square error (RMSE), focusing on the best fitted simulations that result from the simultaneous temperature and salinity optimisation. The RMSE of temperature ranges between 1.00 and

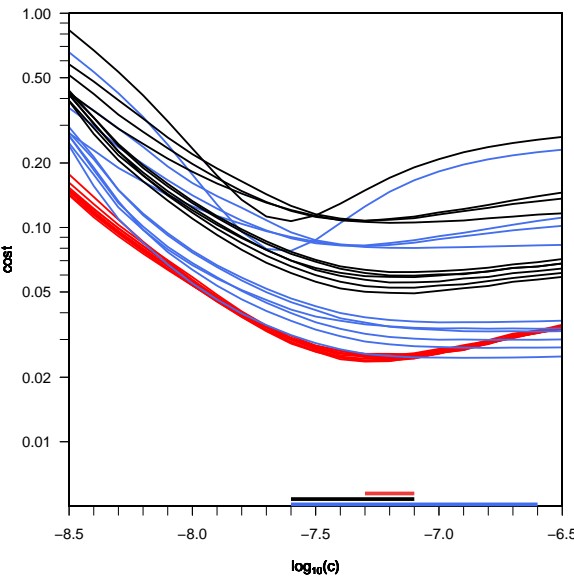

**Figure 3.** Costs of three cost functions depending on parameter $c$ (Eq. 2), for the experiment KOSMOS 2013. Black lines illustrate the cost function $J_{TS}$, taking into account the model-data misfit of temperature and salinity. Red lines illustrate the cost function $J_T$, taking into account only temperature errors; blue lines illustrate the cost function $J_S$, taking into account only salinity errors. The horizontal bars on the x-axis mark the range of all mesocosms' best estimates of $c$ for the three optimisations.

1.09°C, which is much higher than in PeECE III. However, this does not mean that the model performance is worse, since the RMSE should be interpreted in the context of the overall variation in observed temperature. Putting the RMSE of temperature into perspective, it constitutes only 24.2% of the variability (standard deviation) of the observed temperature, which is lower than the corresponding ratio in the PeECE III mesocosms (see 3.1). In contrast, the RMSE of salinity is lower than in PeECE III, ranging from 0.02 to 0.04 PSU, while the ratio of the highest RMSE and the standard deviation of the observed salinity is 40.6% and thus larger than in PeECE III.

## 4  Discussion

In this study, we simulate the temporal-vertical evolution of temperature and salinity in 19 individual mesocosms (PeECE III and KOSMOS 2013), by employing a one-dimensional water column model for mesocosms. The model has two key features that allow for realistic simulations of observed temperature and salinity patterns: 1) a density-based eddy diffusivity parametrisation with one free parameter for optimisation, and 2) a surface correction of heat and freshwater fluxes, based on observed changes in total heat content and integrated water column salinity (see Section 2 for a detailed description). To minimise the model-data misfit, we optimise a diffusivity parametrisation with respect to temperature and salinity. In this





**Table 3.** Best estimates of parameter $c$ and associated combined cost that takes temperature and salinity errors into account, $J_{TS}(c_T)$, $J_{TS}(c_S)$ and $J_{TS}(c_{TS})$, for all mesocosms of the experiment KOSMOS 2013.

| mesocosm | $\log_{10}(c_T)$ | $J_{TS}(c_T)$ | $\log_{10}(c_S)$ | $J_{TS}(c_S)$ | $\log_{10}(c_{TS})$ | $J_{TS}(c_{TS})$ |
|:---:|:---:|:---:|:---:|:---:|:---:|:---:|
| 1 | -7.2 | 0.105 | -7.1 | 0.106 | -7.2 | 0.105 |
| 2 | -7.3 | 0.107 | -7.4 | 0.108 | -7.3 | 0.107 |
| 3 | -7.2 | 0.050 | -6.9 | 0.052 | -7.1 | 0.049 |
| 4 | -7.2 | 0.055 | -6.8 | 0.059 | -7.2 | 0.055 |
| 5 | -7.3 | 0.053 | -6.8 | 0.056 | -7.1 | 0.052 |
| 6 | -7.2 | 0.062 | -7.0 | 0.063 | -7.2 | 0.062 |
| 7 | -7.2 | 0.060 | -6.6 | 0.066 | -7.1 | 0.060 |
| 8 | -7.1 | 0.111 | -7.3 | 0.108 | -7.3 | 0.108 |
| 9 | -7.3 | 0.149 | -7.6 | 0.107 | -7.6 | 0.107 |
| 10 | -7.3 | 0.059 | -6.6 | 0.066 | -7.1 | 0.059 |

**Table 4.** Mean and standard deviation of observed temperature (T) and salinity (S) as well as root mean square errors (RMSE) of simulated temperature and salinity, based on the simulation with $c_{TS}$ as optimal value for parameter $c$. The listed values are for all mesocosms of the experiment KOSMOS 2013.

| mesocosm | $T_{obs}$ (mean ± sd) | RMSE(T) | $S_{obs}$ (mean ± sd) | RMSE(S) | $\log_{10}(c_{TS})$ |
|:---:|:---:|:---:|:---:|:---:|:---:|
| 1 | $7.37 \pm 4.49$ | 0.999 | $29.10 \pm 0.10$ | 0.039 | -7.2 |
| 2 | $7.38 \pm 4.51$ | 1.006 | $29.28 \pm 0.07$ | 0.028 | -7.3 |
| 3 | $7.36 \pm 4.51$ | 0.998 | $29.37 \pm 0.11$ | 0.024 | -7.1 |
| 4 | $7.39 \pm 4.51$ | 0.999 | $29.15 \pm 0.10$ | 0.025 | -7.2 |
| 5 | $7.36 \pm 4.51$ | 0.997 | $29.45 \pm 0.10$ | 0.024 | -7.1 |
| 6 | $7.39 \pm 4.52$ | 1.009 | $29.29 \pm 0.10$ | 0.026 | -7.2 |
| 7 | $7.39 \pm 4.53$ | 1.030 | $29.36 \pm 0.10$ | 0.027 | -7.1 |
| 8 | $7.40 \pm 4.51$ | 1.024 | $29.11 \pm 0.07$ | 0.029 | -7.3 |
| 9 | $7.36 \pm 4.51$ | 1.091 | $29.32 \pm 0.09$ | 0.035 | -7.6 |
| 10 | $7.38 \pm 4.50$ | 1.000 | $29.10 \pm 0.10$ | 0.027 | -7.1 |





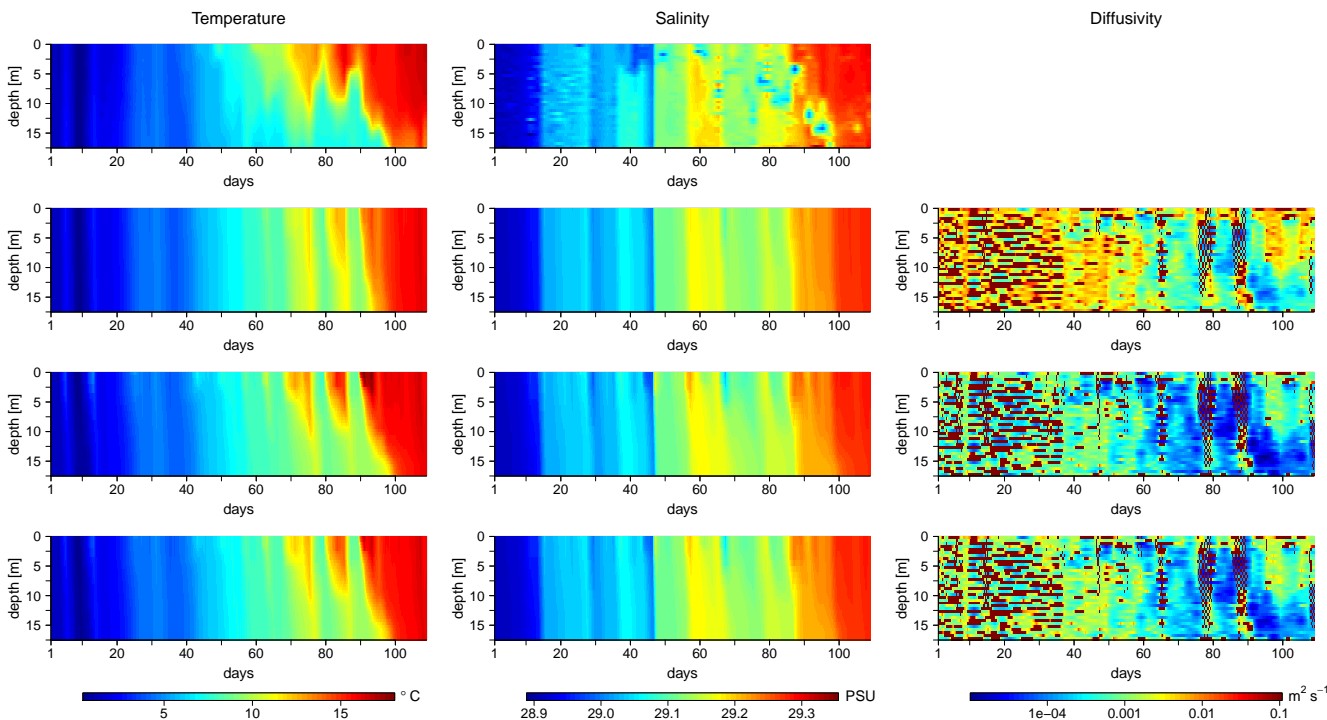

**Figure 4.** Temporal evolution of temperature, salinity, and diffusivity profiles in one exemplary mesocosm of KOSMOS 2013. The first row shows observed temperature and salinity; the following rows show simulated temperature, salinity, and diffusivity for a) the salinity optimisation (second row), b) the temperature optimisation (third row), and c) the optimisation that depends on both temperature and salinity (fourth row). Shown are results for mesocosm #10, figures for the other KOSMOS 2013 mesocosms are provided in the Supplementary Material.

publication, we introduce our model, provide our best estimates of eddy diffusivities for each of the 19 mesocosms (see Supplementary Information) and demonstrate that these estimates lead to simulated patterns of temperature and salinity that closely resemble the observed patterns (see Section 3 and Supplementary Information).

## 4.1 Observed and simulated profiles of temperature and salinity

5   For both mesocosm experiments that we investigated in this study, PeECE III and KOSMOS 2013, our results show that vertical mixing patterns are very heterogeneous in time and space. For the PeECE III experiment, our simulations support the assumption of previous studies (Riebesell et al., 2007; Schulz et al., 2008) that water at the bottom of the PeECE III mesocosms was largely separated from the surface mixed layer during the whole experiment. Within the pycnocline that separates surface and bottom water, the simulated eddy diffusivities are very low, suggesting that tracer exchange between surface and bottom

10  water was mostly inhibited. However, whether tracer diffusion through the pycnocline was still large enough to affect surface





biogeochemistry, can only be detected by a model that includes biogeochemical tracers explicitly. The PeECE III mesocosms show a distinct pattern for each optimisation, due to the sensitive response of the artificial halocline to even slight changes in vertical diffusivities. In the KOSMOS 2013 experiment, the water column was fully mixed in the beginning and subsequently increasingly divided by thermal stratification. Our model is able to capture the very different mixing regimes and it reproduces

the observed temperature and salinity profiles accurately.

In seven out of the 19 simulated mesocosms, either the temperature optimisation or the salinity optimisation result in the same diffusivity estimates as the optimisation that depends on temperature and salinity simultaneously, i.e. in these cases the optimal diffusivities could have been estimated by taking into account only temperature or only salinity, respectively. However, since this is evident only in hindsight, after the different optimisations were conducted, we recommend to estimate diffusivities

always with the simultaneous temperature and salinity optimisation, which provides the optimal diffusivities in any case.

## 4.2  Constant product of dissipation rate and mixing efficiency

For the calculation of eddy diffusivities, we keep the parameter $c$, the product of dissipation rate and mixing efficiency in the eddy diffusivity parametrisation (Eq. 2), constant over time and space. All of our eddy diffusivity estimates are within the range of observed values found in the literature (e.g., Waterhouse et al., 2014; Whalen et al., 2012; Rovelli et al., 2016; Fer,

2009). The overall range of eddy diffusivities within a mesocosm is determined by the parameter $c$, which is optimised for each mesocosm independently. Thereby, the model can account for significant external influences, like local average wind speed or wave intensity. As we demonstrated in Section 3, eddy diffusivities in mesocosms of the KOSMOS 2013 experiment are much higher than those in PeECE III mesocosms, potentially due to differences in local wind speed, geographic position and surrounding currents. We do not recommend to use a constant $c$, if significant seasonal changes in wind speed occur during

the experiment. Both experiment sites we investigated here, Bergen and Gullmar Fjord, are characterised by a stormy winter season and a calmer summer season. Both mesocosm experiments were conducted within the calmer season.

## 4.3  Deriving diffusivities from observations

We derive eddy diffusivities from observed temperature and salinity profiles during the whole simulation period. Alternatively, it would be possible to only initialise the model with observed profiles and subsequently derive diffusivities from simulated

temperature and salinity. However, our approach has the advantage that an accumulation of errors can largely be avoided. If diffusivities were derived from simulated temperature and salinity, even a small error in simulated values would lead to a progression of rapidly increasing errors. Furthermore, using observed temperature and salinity data also has the advantage that the influence of temporary storm events can be taken into account. A storm tends to increase mixing and decrease density gradients. Consequently, our model calculates higher diffusivities, since diffusivities and density gradients are inversely proportional (Eq.

2). This means, even though there is no direct dependency of diffusivities on wind speed, a temporary strong mixing event is automatically induced in the model by using observed temperature and salinity data.





## 4.4 Assumption of uncorrelated temperature and salinity

In the cost function that we use for the diffusivity optimisations, we assume that there is no correlation between salinity and temperature, because temperature and salinity can be positively or negatively correlated, depending on the environmental conditions. A strong increase in temperature at the surface would likely enhance evaporation, thus leading to an increase

in salinity (positive correlation), a pattern that has been observed in the KOSMOS 2013 experiment. Conversely, in case of thermohaline stratification, as present during the PeECE III experiment, warm water with a relatively low salinity lies on top of colder more saline water - mixing of the two water masses would decrease surface temperature and increase surface salinity (negative correlation), and vice versa for water below the pycnocline. Since a combination of these processes could take place any time during the simulation period, temperature and salinity changes are treated as independent in the optimisations.

## 4.5 Limitations of the model

One important assumption in our model is that all changes in integrated water column heat and salinity are caused by heat and fresh water fluxes at the surface. The model does not account for heat fluxes through the mesocosm walls. In reality, heat transfer through the mesocosm walls can be significant when surrounding waters are influenced, e.g., by changing currents that transport colder water to the experiment site, as has been observed during the KOSMOS 2013 experiment (Bach et al., 2016).

Presumably, our model overestimates heat in the lower part of the mesocosms because heat exchange with surrounding waters is unaccounted for. In the model heat can only exit the system after it is transported to the surface by turbulent diffusion. For fresh water fluxes, i.e. salinity changes, this limitation is not critical because even in reality there is no mass flow through mesocosm walls, as long as they are intact. In both experiments, temporary holes were detected but quickly fixed. Whether associated intrusion of surrounding waters had an impact on salinity inside the mesocosms is unknown. In any case, the assumption that

heat (with the exception of penetrative solar heating) and fresh water fluxes occur only at the surface is a potential source for model errors. Another limitation of our model is the lack of nightly convection. Since temperatures were measured only at daytime, it is not possible to account for nightly cooling and related nightly convection. A continuous monitoring of temperature and salinity on site would allow for more exact diffusivity estimates. For future mesocosm experiments, we recommend to put data loggers into each mesocosm that record temperature and salinity continuously throughout the experiment. Thereby, nightly

convection could be taken into account in a mesocosm mixing model and eddy diffusivity estimates could be further improved. Despite these limitations our simulated temperature and salinity profiles match the observations very well, suggesting that daily observations of temperature, salinity and solar radiation are sufficient to simulate the main characteristics of vertical mixing patterns in mesocosms. For the purpose of biogeochemical modelling, the main characteristics like mixed layer depth or pycnocline thickness are most important. Thus, the model we present here can be the basis for future biogeochemical

mesocosm model studies.





### 4.6 Comparison with standard mixing assumptions

Most analyses of mesocosm data assume that there has been either no mixing or full mixing of the water column. To demonstrate that our model is significantly better than these simple assumptions, we compare the cost of our best estimate, $J_{\mathrm{TS}}(c_{\mathrm{TS}})$, with the cost of a simulation without any mixing other than convective mixing, $J_{\mathrm{TS}}(\mathrm{no\ mixing})$, and also with the cost of a simulation with full mixing, $J_{\mathrm{TS}}(\mathrm{full\ mixing})$.

In case of the PeECE III mesocosms, when there is no mixing the cost $J_{\mathrm{TS}}(\mathrm{no\ mixing})$ ranges between 1.30 and 3.17 and the average cost of all mesocosms is 25 times higher than the cost of our temperature and salinity optimisation $J_{\mathrm{TS}}(c_{\mathrm{TS}})$. For full mixing, the cost $J_{\mathrm{TS}}(\mathrm{full\ mixing})$ ranges between 0.53 and 0.55 and is on average six times higher than $J_{\mathrm{TS}}(c_{\mathrm{TS}})$.

In case of the KOSMOS 2013 mesocosms, the cost $J_{\mathrm{TS}}(\mathrm{no\ mixing})$ ranges between 13.57 and 32.72 and is on average 302 times higher than $J_{\mathrm{TS}}(c_{\mathrm{TS}})$. For full mixing, the cost $J_{\mathrm{TS}}(\mathrm{full\ mixing})$ ranges between 0.07 and 0.29 and is on average 47 % higher than $J_{\mathrm{TS}}(c_{\mathrm{TS}})$.

Thus, we can confidently state that our mixing model is able to estimate much more realistic mixing conditions than simple assumptions of no mixing or full mixing and can reproduce observed patterns of temperature and salinity very well.

### 5 Conclusions

This study sets out to simulate the physical processes in the mesocosm experiments PeECE III and KOSMOS 2013 (Schulz et al., 2008; Bach et al., 2016), with a focus on vertical mixing. We present here a one-dimensional mesocosm mixing model that is capable to reproduce the observed temporal evolution of temperature and salinity profiles of both mesocosm experiments despite very different environmental conditions. The density-based diffusivity parametrisation that is used in our model yields plausible patterns of vertical diffusivities that are very heterogeneous over time and space. Optimal diffusivity estimates are achieved when the cost function depends on both temperature and salinity and the misfit between model results and observations is lowest. We provide here our optimal diffusivity estimates for the PeECE III and KOSMOS 2013 experiment (see Supplementary Information). The development of the one-dimensional mixing model we present here can be regarded as an important foundation for future mesocosm model studies. Vertical mixing plays a significant role in mesocosms and has to be taken into account to accurately simulate observed temperature and salinity patterns, as we demonstrate in this study. Since our estimated eddy diffusivities can reproduce both temperature and salinity profiles well, we assume that these diffusivities can be applied to biogeochemical tracers as well. Including both vertical mixing and biogeochemistry in future mesocosm model studies could lead to a deeper understanding of biogeochemical processes in the pelagic, e.g., changes in carbon export due to ocean acidification. We encourage other modellers to work with the diffusivity estimates we provide here or to use our model to calculate diffusivities for any other mesocosm experiment, so that further insights can be gathered from observational data of mesocosm experiments. For future mesocosm experiments, we recommend to measure temperature and salinity profiles continuously with fixed data loggers within the mesocosms, so that the accuracy of diffusivity reanalyses can be further improved.



## 6 Code and data availability

The model code and the best estimates of the eddy diffusivities for all mesocosms of the PeECE III and KOSMOS 2013 experiment are available at doi: 10.1594/PANGAEA.905311 (Mathesius et al., 2019).

*Author contributions.* The mesocosm mixing model was developed by SM together with HD, AO, MS and JG. All programming was done by SM. The cost functions were designed by MS. All calculations and optimisations were carried out by SM. SM wrote the manuscript with important comments from JG, HD, AO and MS.

*Competing interests.* The authors declare that they have no conflict of interest.

*Acknowledgements.* We are grateful to Dr. Kai Schulz and Andrea Ludwig for providing the temperature and salinity CTD data of the PeECE III and KOSMOS 2013 mesocosm experiment, respectively.





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
