# Peer review of "Reanalysis of vertical mixing in mesocosm experiments: PeECE III and KOSMOS 2013"

_Earth System Science Data, 2019_

## Referee Comment (RC1) · Li Gang (Referee) · 21 Jan 2020

Comments on the manuscript entitled "Reanalysis of vertical mixing in mesocosm experiments: PeECE III and KOSMOS 2013 (essd-2019-166)" by Sabine Mathesius et al.

This study developed a one-dimensional mixing model that confidentially reproduced the vertical mixing of the mesocosms of PeECE III and KOSMOS 2013, by comparing the observed and simulated temperature and salinity. Through this model the vertical mixing in mesocosms can be quantitively estimated. I appreciate the work of this study although I honestly am not very familiar with physical oceanography, because it is very useful not only for marine biogeochemical cycles but biological physiology, such as the

photophysiology of phytoplankton.

What I'm mainly concerned are as follows: In the Introduction part, authors just referred the vertical mixing affects the particular matters, nutrients and CO2 flux etc, on Page 2 Lines 12-14. In fact, the model is developed based on the data of PeECE III and KOSMOS 2013 mesocosms studies, one of the very important aspects of which is about the impacts of environmental changes (nutrients, CO2) on phytoplankton physiology. The vertical mixing drives phytoplankton up and down the water column, which affects phytoplankton-experienced light intensity and quality, thus the photophysiology, and ultimately influences marine primary productivity. I think as background the effects of vertical mixing on phytoplankton should be mentioned in the Introduction part as well as the future perspective in the end.

Page 11 Lines 20-21 indicate the temperatures varied from 4.8 to 16.8 oC; while Line 31 shows the mean temperatures ranged between range between 7.36 and 7.40 oC with a standard deviation of 4.49-4.53 oC. I'm confused with the data. If I understand properly, the mean temperature was obtained from all measured values (from surface to bottom, and from start to the end of the experiment). However, according to the first panel of Figure 4 the mean temperature should be close to the medium value of ~10 oC. Moreover, there are big changes of temperatures from 4.8 to 16.8 oC. Averaging the temperature throughout the experiment period missed majority of information when comparing observed and simulated values as described in the text (Page 11 Line 33-35). So, I suggest comparing them day by day, in a temporal scale.

Format the references: Some titles of listed articles are capitalized each word (Page 9, Lines 3, 23 and 31; Page 20, Lines 9 and 14, and Page 21, Line 13), and the remaining ones just capitalized the first word.

Page 21 Line 18, remove the dot.

---

## Referee Comment (RC2) · Anonymous Referee #2 · 24 Jan 2020

The study presents a one-dimensional model for estimation of vertical mixing conditions in mesocosms. The authors offer the model output from two mesocosm experiments as a freely available dataset for further analysis and claim the model to be extendable on other mesocosm experiments.

Alone the focus on the model description makes the suitability of the study to a data-oriented journal like ESSD questionable: It does not seem to find a proper audience here. Under some conditions, however, the model and the generated dataset could attract the attention of other researchers, as if the model would describe a "nontrivial statistical and other methods employed (e.g. to filter, normalize, or convert raw data to primary published data) as well as nontrivial instrumentation or operational methods" (citation from Aims and Scope of ESSD). Unfortunately, the method proposed

here does not fit this definition. Moreover, the background assumptions of the model, its validity for the modeled system, and the usefulness of the generated data appear questionable.

The vertical turbulent diffusion is the main output of the model to be used as an independent variable in research on vertical transport of plankton, nutrients, dissolved gases and other "passive" tracers. The method applied to the estimation of the vertical diffusion is however far from being physically sound or justified for mesocosm conditions. The authors use the Osborn (1980) relationship for diapycnal diffusivity (Eq. 2 of the Discussions paper) as a core for their model, without giving a try to justify this choice for modeling mixing intensity in mesocosms. Several objections against this choice can be raised. I mention here one: if the mesocosm is well-mixed vertically then $N_z = 0$, and $k_z$ in Eq. (2) turns to $\infty$. One could suggest a narrow range of conditions in mesocosms where the Osborn model would still be applicable, but the authors further simplify it by replacing the major variable — the dissipation rate of the kinetic energy of turbulence $\varepsilon$ with a constant $c$ and stating that "...our model assumes that $c$ is constant for all depths and the whole time period of the experiment..." At this point, the baby is thrown out with the bath water. Reformulated in a straightforward way, it means that the vertical turbulent fluxes are explicitly set constant in time and space and decoupled from any forces producing them. The dubious assumption is compensated by fitting of $c$ to the observed changes of temperature/salinity, allowing the model results to eventually agree with observations. Such a workaround apparently loses information about temporal variations in the vertical turbulent fluxes during the fitting period. On the other hand, the measured temperature (salinity) profiles can be directly applied for estimation of $k_z$ by time-space integration of Eq. 9 (without a correction term) with varying spatial integration limits. This straightforward one-equation procedure without loss of temporal variability is known since at least 1925 and is often called the "flux-gradient method" (see e.g. Powell and Jassby 1974 https://doi.org/10.1029/WR010i002p00191 for a review). In this regard, the proposed model is clearly underperforming and has an insufficient predictive power. I encourage the authors to discard the model in favor

of more robust methods and to make instead available the original temperature and salinity data provided with an appropriate description (if not done yet). *The use of the model results for further analysis and application of the model to other mesocosm experiments is not advised*.

A potentially useful model of mesocosm mixing would benefit from paying attention to the mesocosm-related effects on the vertical mixing: reduced solar radiation due to the wall shadowing, heat exchange across the mesocosm walls, reduced wind mixing at the surface. A model incorporating these effects would significantly contribute to analysis of a large number of mesocosm experiments.

---

## Author Comment (AC1) · 11 Apr 2020

We thank the reviewer for her/his constructive comments on our manuscript. We revised our manuscript accordingly. The detailed responses are given below, where R indicates the Referee Comment and A indicates the Author Response.

R: The vertical mixing drives phytoplankton up and down the water column, which affects phytoplankton-experienced light intensity and quality, thus the photophysiology, and ultimately influences marine primary productivity. I think as background the effects of vertical mixing on phytoplankton should be mentioned in the Introduction part as well as the future perspective in the end.

A: We thank the reviewer for this suggestion and added the following sentence to the

[Figure]

Introduction: "By distributing phytoplankton to different depths, vertical mixing also influences the light-exposure and consequently light-sensitive physiological processes of the phytoplankton, such as photo-acclimation, which has an effect on primary production." Furthermore, we now also mention this aspect in the last paragraph of the manuscript: "Including both vertical mixing and biogeochemistry in future mesocosm model studies could lead to a deeper understanding of biogeochemical processes in the pelagic, e.g., changes in carbon export due to ocean acidification as well as changes in primary production due to varying light-exposure and nutrient availability."

R: If I understand properly, the mean temperature was obtained from all measured values (from surface to bottom, and from start to the end of the experiment). However, according to the first panel of Figure 4 the mean temperature should be close to the medium value of 10°C.

A: It is correct that we calculate the mean temperature of each mesocosm by using all data points in time and space. The mean observed temperature in this case (KOSMOS 2013, mesocosm #10) is 7.38°C as stated in Table 4, which is consistent with the observed temperature values depicted in Figure 4.

R: Averaging the temperature throughout the experiment period missed majority of information when comparing observed and simulated values as described in the text (Page 11 Line 33-35). So, I suggest comparing them day by day, in a temporal scale.

A: We agree that much information is lost when discussing only the average values, which is why we thoroughly discussed the temporal differences in simulated and observed temperatures in the Results and Discussion sections (also illustrated in Figs. 2 and 4). We originally considered the introduction of additional figures that compare the differences on a daily basis. However, we realised that these graphics would only provide another perspective, without actually providing new information. We therefore decided to omit these figures.

R: Format the references: Some titles of listed articles are capitalized each word (Page

9, Lines 3, 23 and 31; Page 20, Lines 9 and 14, and Page 21, Line 13), and the remaining ones just capitalized the first word. Page 21 Line 18, remove the dot.

A: We are grateful for this comment and have revised the references accordingly.

---

## Author Comment (AC2) · 11 Apr 2020

We thank the reviewer for his/her comments on our manuscript. We appreciate the reviewer's call for a more complex mixing model that resolves more of the physical processes at play explicitly. However, the problem we see in increasing the model's complexity is an absence of in-situ data to constrain such complex model. Our very simplified approach requires temperature, salinity and solar radiation measurements only. These measurements are typically available for most pelagic mesocosm experiments. Despite its low complexity, our model is already, in its current form, capable of retracing observed temperatures and salinities in the water column of the mesocosm experiments. This is a clear advance compared to the contemporary (standard) way of simulating and interpreting mesocosm experiments, which is to

assume that there is continuous full mixing or no mixing at all - hypotheses that we prove to be inconsistent with the (available temperature and salinity) data in our study. The major aims of this reanalysis data set are to impede such inconsistencies and to provide an expedient alternative to these oversimplified mixing assumptions. Our approach provides valuable insight into the impacts of mixing effects on the outcome of mesocosm experiments. We understand the reviewer's concerns and we agree that credible inferences from mesocosm experiments may call for an even more prudent approach to constraining and simulating the turbulent processes in future mesocosm experiments. Thus, we recommended to install temperature and salinity data loggers inside and outside of each mesocosm in future experiments (page 16, line 23), so that models like ours could be refined for deriving more precise estimates of vertical diffusivities.

In the detailed response below, R indicates the Referee Comment and A indicates the Author Response.

R: Alone the focus on the model description makes the suitability of the study to a data-oriented journal like ESSD questionable: It does not seem to find a proper audience here. Under some conditions, however, the model and the generated dataset could attract the attention of other researchers, as if the model would describe a "nontrivial statistical and other methods employed (e.g. to filter, normalize, or convert raw data to primary published data) as well as nontrivial instrumentation or operational methods" (citation from Aims and Scope of ESSD).

A: Please note that ESSD has hosted papers on data sets from data assimilation methods in the past. Hence, we concluded that our reanalysis does also qualify as being a study where ". . . nontrivial statistical and other methods" are employed. The consideration of mesocosm experimental data for testing and calibrating planktonic ecosystem model components is relevant. Before introducing such components to large-scale or global models, it is meaningful to evaluate their performance against

data of mesocosm experimental data. In previous studies, data-model syntheses or model assessments against mesocosm data were impaired by the simplified mixing assumptions mentioned before. With the diffusivities provided here, we facilitate the assessment of plankton dynamics resolved by different models against the mesocosm data.

R: The method applied to the estimation of the vertical diffusion is however far from being physically sound or justified for mesocosm conditions. The authors use the Osborn (1980) relationship for diapycnal diffusivity (Eq. 2 of the Discussions paper) as a core for their model, without giving a try to justify this choice for modeling mixing intensity in mesocosms.

A: We are thankful for the referee to raise this point and apologise for not having justified the choice of our diffusivity parameterisation. We revised the manuscript accordingly and added the reason why we base our diffusivity parameterisation on Osborn (1980) to the Methods section (page 4, line 4): "Vertical mixing in mesocosms can be influenced by different degrees of stratification of the water column, where the time and depth of stratification (or lack thereof) can be highly variable. Based on the density-dependent diffusivity parametrization of Osborn (1980) that inherently takes stratification-induced inhibition of vertical mixing into account, we parametrize the diffusivity $k_z$ as
$k_z = \ldots$"

R: Several objections against this choice can be raised. I mention here one: if the mesocosm is well-mixed vertically then $N_z = 0$, and $k_z$ in Eq. (2) turns to $\infty$.

A: Considering Eq. (2) alone, it is correct that $k_z$ would turn to $\infty$. To avoid this, the maximum value of diffusivities it set to 0.1 $m^2$ $s^{-1}$, as described in the Methods section of the Discussion paper (page 4, line 18), i.e. even in a well-mixed water column the
diffusivities do not exceed the value 0.1 m$^2$ s$^{-1}$. This maximum value is based on observations of diffusivities in ocean surface waters (e.g., Price et al., 1986).

R: One could suggest a narrow range of conditions in mesocosms where the Osborn model would still be applicable, but the authors further simplify it by replacing the major variable – the dissipation rate of the kinetic energy of turbulence $\varepsilon$ with a constant $c$ [...], it means that the vertical turbulent fluxes are explicitly set constant in time and space and decoupled from any forces producing them.

A: We agree with the reviewer that the dissipation rate is one key variable of the Osborn model. However, the other key variable of this parameterisation is the buoyancy frequency (N), which also resolves the temporal and vertical variability of diffusivities. Consequently, the vertical turbulent fluxes are not constant, despite a constant $c$. We cite here from the Discussion paper's description of the parameter $c$, where we explicitly address this point: "The dissipation rate as well as the mixing efficiency can significantly vary in time and space. However, our model assumes that $c$ is constant for all depths and the whole time period of the experiment. The constant $c$, which is optimised for every single mesocosm independently (see 2.3), can be interpreted as an averaged value within one mesocosm for the entire experiment time. With this simplification, any temporal and vertical variations in the vertical diffusivities k$_z$ are induced by the buoyancy frequency and are inversely proportional to the vertical density gradient." We replaced the previous wording "any differences in the vertical diffusivities" by "any temporal and vertical variations in the vertical diffusivities" for clarification and apologise for the misunderstanding.

R: The dubious assumption is compensated by fitting of $c$ to the observed changes of temperature/salinity, allowing the model results to eventually agree with observations.

A: As already stated above, the constant parameter $c$ can be regarded as an averaged

value of the product of mixing efficiency and dissipation rate for the whole experiment time. In the Discussion section we stress and recommend to apply this pragmatic approach only for periods of time that are short compared to changes in environmental conditions such as water column stratification and the typical strength of turbulent forcing, meaning that seasonal changes in dissipation rate and mixing efficiency do not come into play. The mesocosm experiments typically span time periods of weeks to months. From our analyses we learned that a constant value for $c$ yields good model results, first and foremost being consistent with the temperature and salinity data. In the optimisation of the parameter $c$, we assured that the order of magnitude of values for $c$ is varied within a realistic range of values of the product of mixing efficiency and dissipation rate in mesocosms (personal communication with Marcus Dengler, who measured dissipation rates in pelagic mesocosms in the Baltic sea).

R: Such a workaround apparently loses information about temporal variations in the vertical turbulent fluxes during the fitting period.

A: As discussed above, the temporal variation of the diffusivities is still resolved. Furthermore, the observed density gradients (that are part of our diffusivity parametrisation) are strongly influenced by real-world mixing, with high mixing leading to low density gradients. This in turn results in high diffusivity estimates in our model, consistent with the high real diffusivities. The same is valid for the reversed case. In the manuscript (page 15, line 27) we further describe: "[. . .] the influence of temporary storm events can be taken into account. A storm tends to increase mixing and decrease density gradients. Consequently, our model calculates higher diffusivities, since diffusivities and density gradients are inversely proportional (Eq. 2). This means, even though there is no direct dependency of diffusivities on wind speed, a temporary strong mixing event is automatically induced in the model by using observed temperature and salinity data."
R: On the other hand, the measured temperature (salinity) profiles can be directly applied for estimation of $k_z$ by time-space integration of Eq. 9 (without a correction term) with varying spatial integration limits. This straightforward one-equation procedure without loss of temporal variability is known since at least 1925 and is often called the "flux-gradient method" (see e.g. Powell and Jassby 1974 https://doi.org/10.1029/WR010i002p00191 for a review).

A: Flux-gradient methods are very similar to our approach. The major difference is associated with the introduction of sources or sinks of the simulated tracer, which is required for heat and salinity in mesocosms, where surface fluxes as well as radiative fluxes are relevant. Therefore, the respective correction terms in our approach are absolutely crucial to estimate realistic diffusivities. From the reviewer's suggestion we conclude that we have done a confusing job explaining that we actually do calculate $k_z$ for each depth and hour. More precisely, we calculate N for each depth and hour and multiply it by the optimised parameter $c$ to obtain a realistic $k_z$ for the respective depth and time. In this way we allow for temporal variability of mixing in the mesocosms. We regret the confusion and added a clarifying sentence to the model description, (page 4, line 10): "Both $k_z$ and $N_z$ are calculated for every time step and depth level."

R: In this regard, the proposed model is clearly underperforming and has an insufficient predictive power.

A: As we demonstrate in the paper, our hindcasted temperature and salinity profiles are in good agreement with the observed profiles, even under highly variable mixing regimes in the mesocosms (from full mixing to strong stratification). This indicates a performance clearly superior to the hypotheses of no mixing/full mixing applied in the past. We agree that, given more observations (for which we make recommendations in section 4.5 "Limitations of the model"), more complex and potentially also more reliable models can and should be developed in the future. In this study we present a step forward against studies to come will be benchmarked so that incremental process

towards understanding mesocosms data can be achieved.

R: make instead available the original temperature and salinity data provided with an appropriate description (if not done yet).

A: The original temperature and salinity data are published in Schulz et al. 2008 (PeECE III) and Bach et al. 2016 (KOSMOS 2013). We refer to these publications in our paper, e.g. in the Methods:Data section.

R: A potentially useful model of mesocosm mixing would benefit from paying attention to the mesocosm-related effects on the vertical mixing: reduced solar radiation due to the wall shadowing, heat exchange across the mesocosm walls, reduced wind mixing at the surface. A model incorporating these effects would significantly contribute to analysis of a large number of mesocosm experiments.

A: During the development process of the mesocosm mixing model we examined these (and more) potentially important factors thoroughly and found them to have a negligible effect on the diffusivity estimates. The only exception is the heat flux through the mesocosm walls. Without measured temperature profiles outside each mesocosm, it is impossible to calculate the heat flux through the mesocosm walls, unfortunately. That said, we agree with the reviewer that our approach has limitations because of an uncomprehensive representation of processes such as the ones mentioned by the reviewer. We discuss these caveats in the section "Limitations of the model". We added an additional sentence highlighting the caveats raised by the reviewer (page 16, line 26): "Other processes that are not accounted for include reduction of solar radiation by wall shadowing and the direct effect of wind speed on turbulent mixing." Please note that among our concluding remarks are recommendations to measure temperature profiles also outside each mesocosm in future mesocosm experiments. With such additional data, the mesocosm mixing model could be improved by accounting for the heat flux through the mesocosm walls. Please note also that "reduced wind mixing at the surface" is implicitly included in our model, as the parameter $c$ (which is optimized for each mesocosm) is the product of the wind-speed affected mixing efficiency and dissipation rate. Our model is targeted to make the best out of the available measurements that accompanied the respective mesocosm experiments. We certainly agree that there are more elaborate ways to simulate and parameterise turbulent processes in the water column.

References:

Bach, L. T., Taucher, J., Boxhammer, T., Ludwig, A., Kristineberg KOSMOS Consortium, Achterberg, E. P., ... Czerny, J. (2016). Influence of ocean acidification on a natural winter-to-summer plankton succession: First insights from a long-term mesocosm study draw attention to periods of low nutrient concentrations. PloS one, 11(8), e0159068.

Price, J. F., Weller, R. A., Pinkel, R. (1986). Diurnal cycling: Observations and models of the upper ocean response to diurnal heating, cooling, and wind mixing. Journal of Geophysical Research: Oceans, 91(C7), 8411-8427.

Schulz, K. G., Riebesell, U., Bellerby, R. G. J., Biswas, H., Meyerhöfer, M., Müller, M. N., ... Zöllner, E. (2008). Build-up and decline of organic matter during PeECE III.